# Surface Comparison of Three Different Commercial Custom-Made Titanium Meshes Produced by SLM for Dental Applications

**DOI:** 10.3390/ma13092177

**Published:** 2020-05-08

**Authors:** Nuno Cruz, Maria Inês Martins, José Domingos Santos, Javier Gil Mur, João Paulo Tondela

**Affiliations:** 1Faculty of Dentistry, Universitat Internacional de Catalunya, 08017 Barcelona, Spain; xavier.gil@uic.cat; 2Faculty of Engineering, University of Porto (FEUP), 4200-465 Porto, Portugal; up201305982@fe.up.pt; 3REQUIMTE-LAQV, Department of Metallurgical Engineering and Materials, Faculty of Engineering, University of Porto (FEUP), 4200-465 Porto, Portugal; jdsantos@fe.up.pt; 4CIROS from the Faculty of Medicine, University of Coimbra, 3004-504 Coimbra, Portugal; jtondela@fmed.uc.pt

**Keywords:** biomaterial, bone regeneration, titanium mesh, 3D printing, surface properties, roughness

## Abstract

The use of individualized titanium meshes has been referred to in scientific literature since 2011. There are many advantages to its use, however, the main complications are related to early or late exposures. As some aspects such as its surface properties have been pointed out to influence the soft tissue response, this study was designed to compare the surface characteristics of three commercially available individualized titanium meshes between them and according to the manufacturer’s specifications. The results from the scanning electron microscopy, energy-dispersive X-ray spectroscopy, X-ray diffraction and the contact profilometry measurements were analyzed and cross-checked. It was discovered that, the BoneEasy’s post-processing superficial treatment was more refined, as it delivers the mesh with the lowest Ra value, 0.61 ± 0.14 µm, due to the applied electropolishing. On the other hand, the Yxoss CBR^®^ mesh from ReOss^®^ was sandblasted, presenting an extremely rough surface with a Ra of 6.59 ± 0.76 µm.

## 1. Introduction

Oral and maxillofacial reconstructive attempts are traceable back to as early as the Egyptian and South-Central American cultures. Obviously, over the centuries and with the development of material, biological and medical sciences, techniques that fulfill both the functional and esthetical requirements of dental implants have arose [1].

Currently, fixed prosthetics solutions sustained by dental implants for oral rehabilitation no longer consist in their simple placement following the existing bone anatomy. Instead, the current trend demands for individualized solutions and treatment plans that start way before and go far beyond the simple computed tomography (CT)/cone beam computed tomography (CBCT) bone defect scan analysis.

Many patients suffer from horizontal or vertical bone deficiency, especially in cases of long-lasting edentulous ridges and bone defects, which are frequently caused by trauma or bone pathology. Thus, for prosthetic-driven procedures, the existent resorbed alveolar bone is often not enough for dental implant placement and even when possible, it frequently jeopardizes the successful outcome of an optimized implant placement. In order to prevent such complications and to achieve an appropriate positioning of dental implants, several augmentation strategies have been developed with the purpose of favoring the new bone’s growth. Some of these techniques include alveolar distraction osteogenesis, block bone graft and guided bone regeneration [2,3].

One of the most familiar and most commonly used strategies, guided bone regeneration (GBR), resorts to a barrier membrane to isolate the growth of soft tissue while promoting the bone tissue growth as a priority [4]. However, especially for large bone defects, the desired bone shape and volume are hard to maintain throughout the entire GBR healing period. Furthermore, graft material displacement and compression during the post-operative period have been cited as relevant phenomena [5].

Since its introduction in 1969, the titanium mesh has received profound attention and has been extensively used for the reconstruction of oral and maxillofacial bone defects; its intensive use is due to its favorable characteristics [6,7]. The titanium mesh is rigid enough, being able to control bone shape and volume, a basic prerequisite for any bone regeneration process, and its pores play an important role both in enabling the vascular supply from the overlaying periosteum to the grafted defect and in improving tissue integrity [8].

However, despite the tremendous potential of the titanium mesh, obviously some limitations were also recorded; for the application of the conventional mesh, manual shaping through cutting, bending and trimming is required. These processes are very manually challenging, time-consuming and highly influence the overall regenerative outcome [9]. Furthermore, the corners and edges of the bended and cut meshes can cause severe gingiva damage and expose the mesh’s site [10].

Fortunately, in recent years, the development of personalized rapid prototyping medical devices based on the digital imaging and communications in medicine (DICOM) files provided by CT/CBCT scans, has deeply intensified [11]. Based on the patient’s bone defect and resorting to computer aided design (CAD) software, it is possible to design medical devices with the intent of recreating the lost tridimensional bone anatomy. Furthermore, the virtual design can be physically produced by a recurring tridimensional (3D) printing technique.

Without a doubt, individualized titanium meshes for bone regeneration are an excellent example of a medical device whose quality has greatly benefited from these technological advances. In fact, resorting to selective laser melting (SLM) and a powder bed fusion 3D printing process [12], custom made meshes are already being produced worldwide. The personalized manufacture of titanium meshes through the digital modelling and 3D printing integration enables the accurate reconstruction of the bone’s volume and position, promoting an optimal fit between the mesh and the anatomical shape as well as grants the opportunity for the procedure to be planned in advance. In addition, by avoiding manual shaping and the pruning of the implantable device in the moment of application, the procedure’s duration can be greatly shortened [3] and the medical outcome of the surgery can be notably enhanced [13].

Titanium is a well-established choice as a material for use in biomedical applications. Its remarkable biocompatibility properties are due to the existence of a superficial passive oxide layer that is formed by the electromechanical oxidation of the material and delivers the titanium’s excellent resistance to corrosion in combination with its excellent chemical inertness [14]. In vitro studies have implied that the negatively charged and hydrophilic TiO_2_ layer is, in fact, the key factor for the overall biocompatibility as it regulates the protein adsorption [15]. For the particular case of the dentistry, countless studies have already been conducted in order to guarantee the implantation safety. Usually, no inflammatory response signs are found in the oral tissue adjacent to titanium implants, however, it is important to note that for some patients, hypersensitivity can be induced [16].

Regardless of the production technique, either by conventional methods or by rapid prototyping, as for all implantable devices, it is important to control the meshes’ characteristics to optimize its biological performance [1]. The inherent stiffness of titanium meshes can be responsible for causing irritation to the soft tissue, and properties such as mechanical strength, that deeply influence the meshes’ use success, are affected by the thickness of the material and pore characteristics, size and number. More specifically, the surface properties of the biomaterial highly direct the interactions at the implant–cell interface [17]. These properties that range from physical to chemical features, including surface topography and chemical composition, are usually dictated by the superficial treatments applied whose current importance is well established [18]. Physicians favor the use of implantable devices that have undergone surface treatments that improve the success rate, accelerating the osseointegration mechanism [18,19]. Furthermore, it has already been reported that dental implants without surface treatments are associated with higher healing times when compared with the treated ones [19]. Surface topography and roughness are some of the aspects that can be easily manipulated by resorting to post-production surface treatments and that play an important role in the determination of cellular response, influencing adhesion, adsorption and differentiation [17,19]. High roughness degrees represent a major risk as the ionic leakage from the material can increase [20] and the bacterial adhesion is facilitated, intensifying the possibility of implant failure [21]. Smooth surfaces are able to slow down the biological processes at the interface, keeping the titanium oxidized layer properties unaffected for longer time periods [19]; the associated correct micro- and nano-roughness level can stimulate osteoblast differentiation, proliferation and production of both matrix and local growth factors [22]. Furthermore, changes in roughness correlate with selective protein adsorption, collagen synthesis and the maturation of chondrocytes, which all significantly influence the implant’s osseointegration [23].

Some of the frequently used treatment techniques include sandblasting, acid etching and electropolishing; each one imprinting unique topographic features on the treated surfaces [19]. By projecting pressurized particles, the sandblasting treatment delivers titanium surfaces with roughness values highly superior relative to the ones in a controlled polishing technique, and is responsible for the introduction of contaminants into the surface. Regarding the acid etching, the treatment with strong acids cleans the metal substrate and delivers homogeneous roughness attributes throughout the entire surface [24]. In turn, electropolishing is a electrochemical process that delivers titanium surfaces with a bright, clean and smooth appearance, through the removal of a thin top layer of the material [25,26]. For the particular case of the SLM-based production of titanium constructs, post-production processes are fundamental and consist in both a thermal treatment and surface treatments, required to remove the raw metal particles that remain bonded to the manufactured piece. These particles, that stick to the structure due to the thermal diffusion associated to the temperature difference between the solidified material and the loose powder, must be removed as after implantation, as they can be released into the surrounding biological environment, possibly leading to inflammation, and are related to the loss of adequate mechanical properties such as fatigue resistance [27].

Even though the meshes characteristics should be deeply tailored and controlled in order to deliver the best possible clinical outcome, it is important to note that the biological progress associated to its use also greatly depends on the correct diagnosis and clinical indication as well as on the host characteristics themselves, such as medical history, the location and dimension of the bone defect and the type of residual bone, among others.

The present study intended to study the surface properties of three different commercially available individualized titanium meshes produced by SLM. The samples’ morphological surface analysis was carried out by scanning electron microscopy (SEM) and the elemental analysis for chemical characterization was fulfilled resorting to an energy-dispersive X-ray spectroscopy (EDS). X-ray diffraction (XRD) was used to discern detailed information about the chemical structure of the materials and a contact profilometry measurement took place to evaluate the meshes’ roughness.

## 2. Materials and Methods

The implantable devices used for this study were custom-made titanium meshes, produced in order to fit perfectly to each patient’s specific needs. The acquired meshes analyzed were Mesh4U from BoneEasy (Arada, Ovar, Portugal), Yxoss CBR^®^ mesh from ReOss^®^ (Filderstadt, Esslingen, Germany) and 3D-MESH from BTK (Dueville, Vicenza, Italy), selected as they were the ones available in the European market with more expressive presence.

To evaluate the main design features of each mesh, the dimensions were determined by measuring in triplicates using a digital caliper.

To evaluate the material’s structure and composition, XRD analyses were carried out resorting to the Bruker D8 Discover equipment (Bruker, Billerica, Massachusetts, USA). The XRD acquisition was performed in the 5°–80° 2θ degree range with a 0.04° step size and an acquisition time corresponding to 1 s per step.

The meshes’ superficial morphology was analyzed trough scanning electron microscopy. The samples were attached to aluminum supports using carbon tape and the analysis was performed resorting to the Quanta 400 FEG ESEM/EDAX Genesis X4M (Thermo Fisher Scientific, Hillsboro, OR, USA): a high resolution (Schottky) environmental scanning electron microscope with X-ray microanalysis and electron backscattered diffraction analysis (Thermo Fisher Scientific, Hillsboro, OR, USA). Furthermore, resorting to the same system, the samples were characterized using energy-dispersive X-ray spectroscopy.

The surface’s profile was analyzed in triplicate, through a contact profilometry measurement, using the Hommel Werk LV-50 equipped with a 5μm radius TK pointer (Hommelwerke Co, Villingen-Schwenningen, Schwarzwald-Baar, Germany). The data acquired were processed by the application of a Gaussian filter in order to isolate roughness from the waviness and shapes of the samples.

## 3. Results

Foremostly, a simple morphological evaluation took place as all the samples displayed themselves with unique identities. The BoneEasy (Figure 1a) and the BTK (Figure 1c) meshes shared similarities as they closely resemble dense plates where circular apertures were planted. In this way, the main feature to evaluate corresponded to the pore diameter. On the other hand, the Yxoss CBR^®^ mesh from ReOss^®^ (Figure 1b) presented a maze-like shaped surface, composed of two distinct coordinating elements that together formed a repetitive pattern: regular circular pores intercalated with longer apertures with a peanut-like shape.

The acquired results of the major design features of each mesh, including both the regular pores’ diameter and the bigger structures’ lengths, are represented in Table 1 by the calculated arithmetic mean of three measurements.

To study the material composition of each individual mesh, a superficial X-ray diffraction analysis and an energy-dispersive X-ray spectroscopy was carried out. The experimentally obtained results are presented in Figure 2 and Figure 3, respectively.

A first preliminary analysis revealed that all the implantable meshes displayed a very similar XRD diffraction pattern, suggesting that their structural composition was identical (see Figure 2). 

When inspecting the EDS attainments (Figure 3), the same overall chemical identity also seemed to be shared as the presented spectra displayed, in a generalized way, the same emission lines. It appears evident that the major chemical dominance was granted by the titanium presence.

As for the topographical assessment of the meshes’ surfaces, both a morphological analysis resorting to electronic microscopy and a roughness investigation based in the contact profilometry results were carried out.

The scanning electron microscopy images of the implants’ surfaces are presented in Figure 4.

For the surface texture analysis, high magnifications levels were adopted. The microscopic findings revealed that, from all the analyzed meshes, Mesh4U (Figure 4a) was the one that presented an overall more polished appearance with a homogenous and smooth presentation. However, long patterned surface sulci were easily identified throughout the sample’s surface.

In contrast, the Yxoss CBR^®^ mesh (Figure 4b) was without a doubt the sample that presented the most irregular surface. Smooth areas could not be identified as the totality of the surface consisted in very irregular sharp projections and depressions.

Finally, being neither the toughest nor the smoothest, the BTK mesh’s surface (Figure 4c) displayed a binary topographic expression; both the flat areas and rough cavities could easily be found on the analyzed sample surface.

Surprisingly, while inspecting the overall topographical features of the meshes, some defects were identified. For both the ReOss^®^’s and BTK’s samples, it was possible to pinpoint unexpected randomly dispersed irregular structures that appeared to be embedded in the surfaces. Close-ups on these details are presented in Figure 5. It was noted that in the Mesh4U sample, no such defects were identified.

In order to inspect these features’ compositions, energy-dispersive X-ray spectroscopy assessments were conducted targeting the specific areas of interest which are also highlighted in Figure 4. The corresponding EDS results are presented in Figure 6 and Figure 7.

The first point to note was that, apart from the morphological divergence, the chemical composition of the defects was very diverse, either when comparing the different structures within the same sample or when considering different samples.

Considering the Yxoss CBR^®^ defect (Figure 5a), two distinct types of infiltrations could be found. The features like the one marked as interest area 1 possessed bigger dimensions and their composition was mainly granted by the presence of aluminum and oxygen (Figure 6a). These artifacts were surrounded by multiple cracks where the smaller contaminations, in interest area 2, were built-in, presenting a granular form. The EDS results (Figure 6b) revealed that these masses were composed, essentially, of silicon.

As for the BTK mesh (Figure 5b), two different phases could also be found when analyzing the surface defect. The first one, area of interest 1, was easily identified throughout the acquired microscopic images due to its whitish and shiny appearance. Even though its presence was very obvious, its dimensions were diminished and the EDS results, presented in Figure 7a, revealed the presence of some metallic elements such as chromium, iron and manganese. Finally, the BTK interest area 2 could be described as large structures with irregular limits and their chemical identity, disclosed by the EDS analysis (Figure 7b), only registered a small deviation from the expected base composition due to the existence of carbon and oxygen peaks.

Returning to the lined up topographical evaluation and, although the microscopic analyses allowed a superficial qualitative assessment of the overall surface roughness of the meshes, for a more quantitative interpretation, contact profilometry measurements were performed. Figure 8 displays, for each surface, the results of the roughness profile monitoring in triplicate, for a 1200 µm sampling length.

A simple plot evaluation corroborated with the already achieved inference that the different meshes presented considerably distinct topographies. In fact, while the BoneEasy’s mesh presented a very regular surface with minimal variation in the profile’s height, the surface of the Yxoss CBR^®^ mesh from ReOss^®^ displayed extremely profound profile variations, as substantially high and low peaks were registered. Considering the BTK 3D-MESH, it was possible to predicate that, even though the surface was not so uniform as the Mesh4U, it came closer to it than to the extremely irregular Yxoss CBR^®^ mesh.

Obviously, the number of surface parameters one can evaluate is very large and their wide range allows a full characterization of each particular surface feature. In the present study, the topographic assessment rested only on amplitude parameters, namely, the average roughness (Ra) and the root mean square deviation (Rq or RMS). The acquired Ra and Rq values, resulting from each triplicates’ average, are presented in Table 2 among with the respective squared deviations, to double check the information presented in the roughness profiles.

Thus, the BoneEasy’s mesh was the one that exhibited the lowest roughness values when analyzing either the Ra, 0.61 µm and the Rq value, 0.73 µm. Significant attention should fall on the ReOss^®^ mesh which presented extremely high values of Ra and Rq, 6.59 µm and 8.39 µm respectively. Once again, as an intermediate between the other two meshes but much closer to the Mesh4U surface characteristics, the Ra parameter of the BTK mesh was 1.63 µm while the Rq was 2.08 µm.

## 4. Discussion

The overall success of a reconstructive dental procedure, encompassing an individual mesh introduction, relies profoundly on the physical-chemical properties of the implant’s surface [19].

Thus, the first question that should be clarified is the composition of each mesh. Presently, without a doubt, commercially pure titanium and titanium–aluminum–vanadium (Ti–Al–V) alloys have established themselves as the prime choice materials for implants in dental applications [28,29] due to their remarkable biocompatibility, which is associated to the formation of a stable oxide layer on their surfaces and their favorable mechanical properties [1]. In fact, the surface oxide layer has been described as one of the main features that controls the titanium implant’s integration in bone [30] since it regulates cellular attachment, highly influencing cell shape and function [31]. Even though the link between the contaminations’ presence and the overall failure of the implant has not been fully explained, the lack of clinical success is often linked to the changes in the biocompatibility properties of this surface that may occur due to the presence of contaminations during the autoclaving process and to the contaminations’ release from the surface, enhancing the inflammatory response [30]. This superficial passive oxide layer is responsible for delivering the titanium’s distinctive corrosion resistance [14] and for that reason, alterations on its chemical identity could cause the dissolution of the implant [31], compromising its mechanical properties [14].

Titanium and titanium-based materials are usually composed by a combination of two distinct crystallographic phases: a hexagonal close packed alpha (α) phase and a body-centered cubic beta (β) phase. However, the two phases coexist in a balance that is determined by the thermal experience of the material or the presence of alloying elements [1,32]. For the case of biomedical applications, the most commonly used alloying elements are aluminum and vanadium, used in the exact proportions that give rise to the well known Ti–6Al–4V alloy [1]; these elements are responsible for stabilizing the titanium’s α and β phases, respectively [32].

The analyzed meshes are no exception as their composition, qualitatively revealed by the EDS results, determined that in fact these meshes’ raw material fell under the above described categories. However, the distinction between the two possible metal substrates, pure titanium (medical grade 4) or a Ti–Al–V alloy (medical grade 5) is extremely difficult; the titanium element displays two major emission lines, of which the secondary one, being around 5 keV, overlaps with the vanadium one [33]. Thus, for a clearer discrimination it is necessary to resort to another characterization technique.

In fact, the XRD outcome suggested that the chemical identities of all three meshes might be the same as the different diffractograms appeared to coincide. While commercially pure titanium consists entirely of the alpha crystallographic phase, the Ti–6Al–4V alloy’s structure comprises both the alpha and beta phase [29]. In this way, the XRD identification of the β phase should allow the distinction between pure titanium and titanium alloy samples. However, since the Bragg reflections relative to the β phase were weaker than the α phase’s ones, they were easily overlapped [32]. Thus, the X-ray diffraction results were also not reliable for crystallographic phase identification and a more adequate technique, such as a metallographic analysis, should be minded for further inquiry in order to provide a clear distinction between pure titanium and titanium alloy samples. However, it is clear that the collected spectra matched, indeed, either to pure titanium or to a Ti–6Al–4V alloy material as the identified main diffraction peaks, at 35°, 38° and 40°, matched to the ones reported for this type of material [34,35].

While the choice of material is extremely important when designing an implantable medical device, the final surface properties also play a key role in the success of the overall process, determining its interactions with the surrounding host tissue [36,37]. In fact, the superficial finishes highly affect the cell adhesion, spreading and differentiation that, in turn, are directly involved in the osseointegration mechanism [19,36].

Roughness is without a doubt one of the main aspects to mind since the implant’s texture highly influences the tissue response [38]. In the past, smooth dental implant surfaces were desired [39], however, with the current awareness that completely smooth surfaces do not allow tissue adhesion, possibly leading to body fluid accumulation and inflammation [40], the prevailing trend points towards the use of moderately rough implant surfaces [39].

With this in mind, the meshes’ surface profiles were examined; the microtopographic features of the implant surface (peaks, valleys and protrusions) are an essential factor in the biological response and the configuration of the bone–implant interface [41]. The Ra parameter, the roughness average, corresponds to the average distance from the profile to the mean line over the length of sampling and is not susceptible to the difference of peaks and valleys. In turn, the Rq value, the root mean square deviation, is the square root of the square of the deviation of the profile from the mean line and in this way, is more sensitive to peaks and valleys. Being associated with measuring instruments that grant higher repeatability and that have been more commonly adopted in monitoring production processes, Ra is more relevant for further discussion [42,43]. It has already been recorded that osteoblasts display greater affinity for the implant surface when it presents a microroughness degree associated to a Ra value of 0.5 µm [44,45]. In this way, it is possible to conclude that the BoneEasy’s mesh was the one that had a surface associated Ra value closer to the recorded target and that the one from ReOss^®^ was the one that diverged the most both from the reported desired roughness characteristics and also from the other analyzed meshes’ superficial profiles.

Having the contact profilometry results analyzed, it is important to match them with the visual assessment from the SEM results. Regarding the Mesh4U sample, one could easily accept the low Ra value obtained since the mesh’s surface without a doubt presented a highly smooth finish. On the other hand, for the higher Ra surface on the Yxoss CBR^®^ mesh from ReOss^®^, delivered microscopic images indeed confirmed the rougher nature of the surface; the irregular clusters from a recognizable distinct nature, along with some cracks and projections, justified the higher variations of the profile distance to a mean line that reflected on the obtained Ra value. However, the major discussion topic arose when evaluating the results from the BTK 3D-MESH. Even though the Ra and Rq values of this surface were slightly higher than the ones of the BoneEasy sample, the registered deviation seemed to not corroborate with the evident topographical differences between the samples; such an irregularity degree of the BTK surface would translate into higher Ra and Rq when comparing to the BoneEasy’s one. A possible explanation rests on the contact profilometry acquisition method. While the first two samples, Mesh4U and Yxoss CBR^®^, were analyzed by a 4.8 mm sampling length and then processed with a 0.8 mm Gaussian filter, due the intricate design and lack of continuous superficial area on the BTK’s mesh, only a short and inadequate 1.5 mm length was covered and the acquired data were refined with a 0.25 mm Gaussian filter. In this way, the results may not be reliable and should not be straightforwardly compared with the ones of the other studied meshes. For a more solid evaluation, additional testing and resorting to alternative analysis methods that would not be compromised by the meshes’ complex shape should be executed.

Proceeding with the morphological assessment, the smooth finish of the BoneEasy’s mesh’s surface was a strong indicator of the electropolishing technique applied to eliminate sharp edges, cracks and pits. The electropolishing process delivers pieces with a very intense natural metallic shine, as recognizable trough a preliminary unaided eye evaluation of the mesh, with higher corrosion resistance and clean from imperfections [26]. It is possible to hypothesize that the long imprints on the BoneEasy’s mesh’s surface corresponded to residual polishing contours.

Regarding the artefacts found on the surface of the Yxoss CBR^®^ sample, due to the existence of the deep cracks that surrounded them, it was conjectured that these features must have been incorporated after the first production steps of the mesh. The EDS analysis revealed that the embedded residues with bigger dimensions were primarily constituted of aluminum and oxygen. It was presumed that these alterations in the surface were induced by the post-production alumina (Al_2_O_3_) sandblasting process, used to improve the surface’s texture. Even though this surface treatment is widely used, it is also known that it may introduce impurities originating from the blasting grits which, in turn, being extremely difficult to remove, may negatively affect the biocompatibility and bone formation beneath it. Furthermore, these alumina contaminations are known to be able to, when in a physiological environment, weaken the implant’s corrosion resistance, compromising the mechanical properties [46,47,48], and may be responsible for debilitating bone formation, constraining regular bone deposition and mineralization [30]. An alternative that could be explored consists of the use of TiO_2_ blasting particles that would not introduce foreign elements to the surface chemistry [30]. Moreover, the presence of the small silicon structures embedded into the alumina surrounding the cracks could be associated with the same sandblasting process; it is usual to resort to silicon for the production of sandblasters nozzles, with the absorption and shock protection intent in mind. However, due to the high aggressiveness of the used blast media, it is possible for the material to start cracking or even shatter with use [49]. In this way, one can speculate that, during the sandblasting process of the Yxoss CBR^®^ mesh, essential for the post-production procedure, a silicon nozzle already over its predicted lifespan or possessing some deficiency was used, causing the release of some of its fragments that, in turn, lodged in the sandblasted surface.

The BTK 3D-MESH did also present signs of an electropolishing finish, with areas of clear lower roughness and polishing traces. However, the process must have been flawed since the surface was also rich in non-polished pits. Concerning the detected contaminations, the most relevant one was perhaps the one marked as interest area 1. These small metallic structures were easily found all over the mesh’s surface and the EDS results suggested that these contaminations were, in fact, evidences that another production raw material, more specifically a stainless steel powder, was already being used in the equipment associated to the current production process. In fact, it was conjectured that the present material was a cobalt–chromium (Cr–Co) alloy, extensively used for medical applications, more particularly of great importance in the dental implant field due to the Cr presence; it is believed to deliver favorable biological and mechanical characteristics [50]. Furthermore, Cr–Co alloys are easily processed and sterilized and present high corrosion resistance [1,50]. Even though implants having these alloys as a chemical foundation have been successfully used over the years for clinical dentistry restorations, they have turned obsolete and gradually been replaced by titanium and titanium-based materials [1,50]. Additionally, these materials have recently lost their trustworthiness and their use is now non-advisable, as many reports have been published, including by the European Chemicals Agency, exposing cobalt’s inherent toxicity when released in the biological environment during corrosion [50,51]. Thus, even though the relative amount of this contamination is doubtlessly very reduced, it is still important to emphasize its presence as it reveals some weaknesses in the production process of the BTK mesh.

## 5. Conclusions

There are many commercially available titanium bone regeneration meshes that, due to their personalized production methods resorting to modelling and 3D printing, perfectly fit to the patient’s defect, greatly improving the reconstructive process outcome. In this work, three of these medical devices were analyzed, more specifically at the superficial properties level which is known for highly influencing the surrounding cellular response.

Particular interest was given to the roughness studies. The BoneEasy’s mesh was the one that presented the lowest Ra value and was the mesh that got the closest to the reported optimal roughness degree that enhances the osteoblasts’ affinity to the surface, reported as 0.5 µm. The reported differences between the surfaces were due to the divergent post-production superficial treatments applied. Mesh4U endured an electropolishing treatment of high quality that was able to deliver flawless smooth surfaces. Contrastingly, the BTK produced mesh also withstood the same polishing process but its surface displayed countless non-polished pits, exposing the less perfect treatment application. In addition, evidences of stainless steel contamination were found on this mesh surface. The Yxoss CBR^®^ mesh suffered a sandblasting treatment that, apart from introducing alumina and silicon impurities onto the surface, was responsible for the very high roughness values that were reported.

## Figures and Tables

**Figure 1 materials-13-02177-f001:**
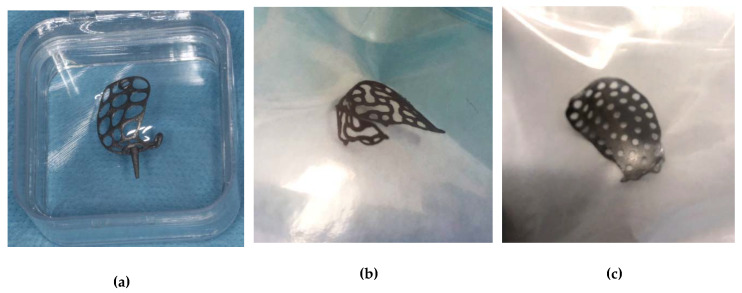
Acquired meshes as received from the manufacturers: (**a**) Mesh4U from BoneEasy; (**b**) Yxoss CBR^®^ mesh from ReOss^®^; (**c**) 3D-MESH from BTK.

**Figure 2 materials-13-02177-f002:**
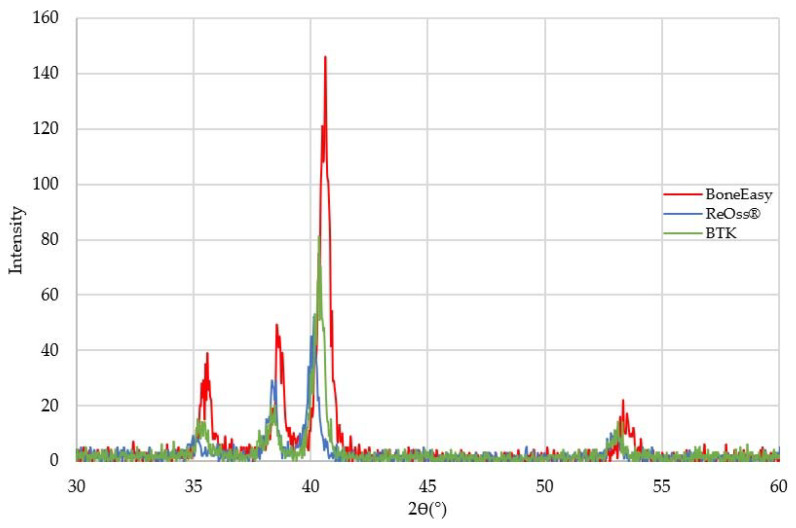
Combined diffractogram of the three meshes analyzed through XRD.

**Figure 3 materials-13-02177-f003:**
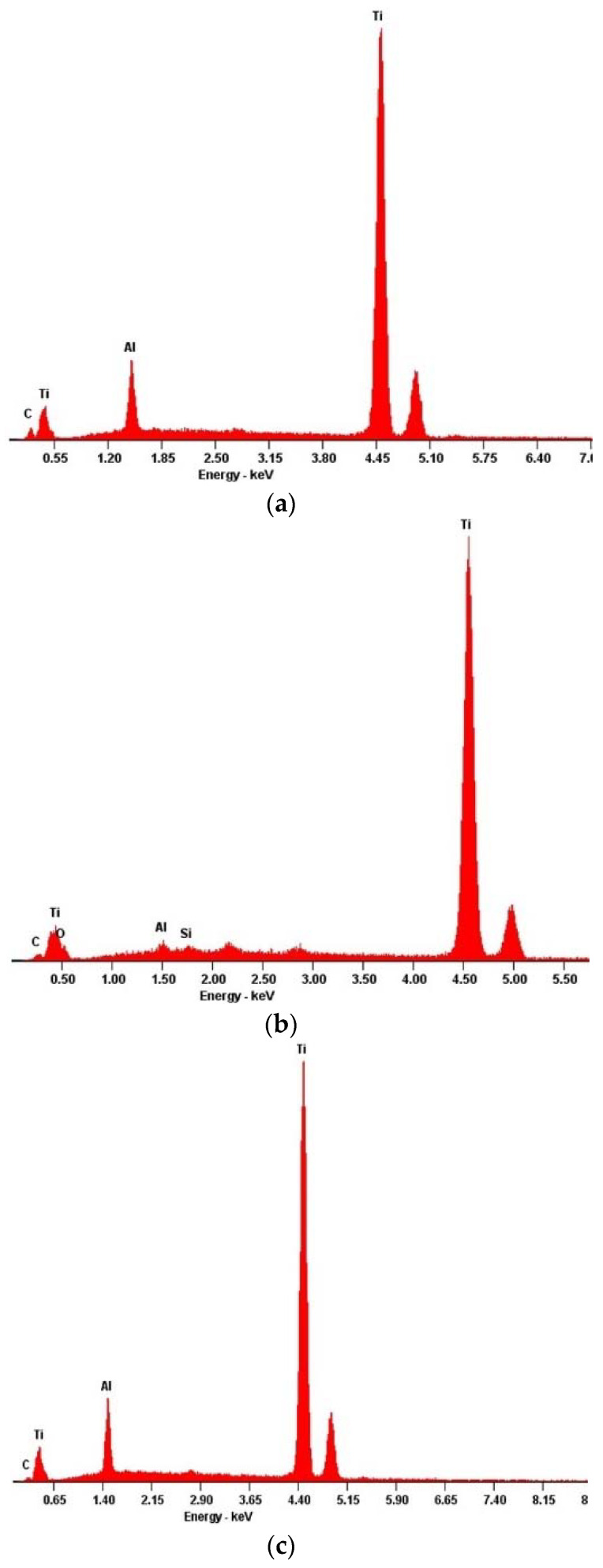
Energy dispersive X-ray spectroscopy results for the different samples’ defects: (**a**) Mesh4U from BoneEasy; (**b**) Yxoss CBR^®^ mesh from ReOss^®^; (**c**) 3D-MESH from BTK.

**Figure 4 materials-13-02177-f004:**
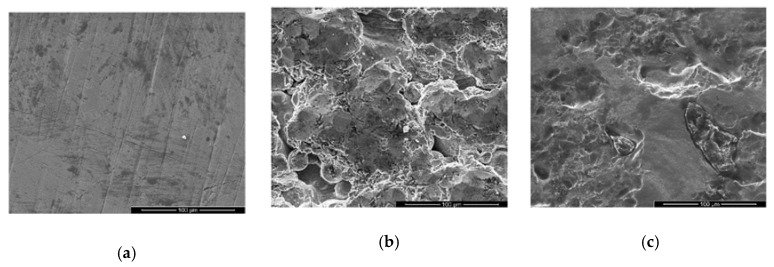
Scanning electron microscopy images for the morphology assessment of the different meshes’ surfaces with 1000x magnification and in secondary electron mode: (**a**) Mesh4U from BoneEasy; (**b**) Yxoss CBR^®^ mesh from ReOss^®^; (**c**) 3D-MESH from BTK.

**Figure 5 materials-13-02177-f005:**
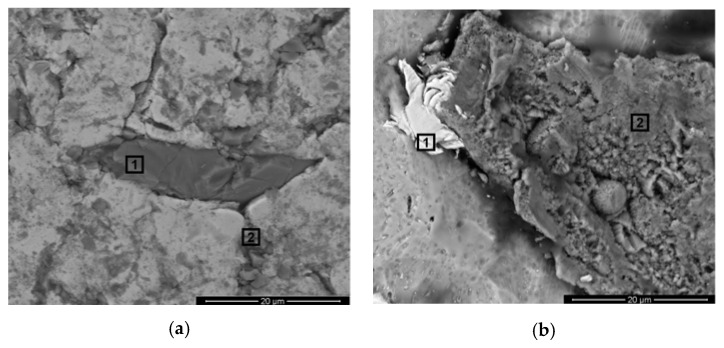
Scanning electron microscopy images with 5000x magnification and in back scattering electron mode for the morphology evaluation of the identified surface defects. Interest areas for further analysis are signalized: (**a**) Yxoss CBR^®^ from ReOss^®^; (**b**) 3D-MESH from BTK.

**Figure 6 materials-13-02177-f006:**
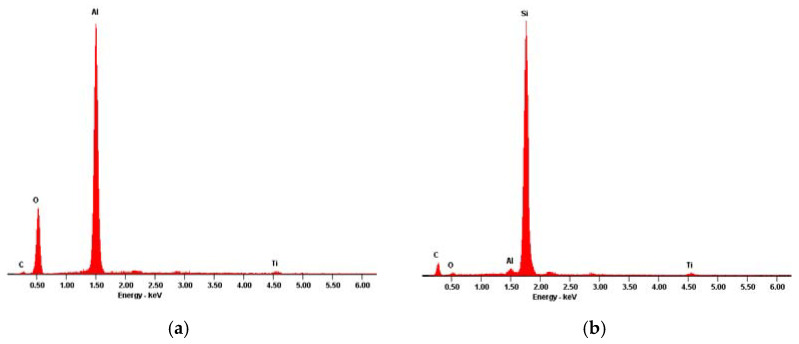
Energy dispersive X-ray spectroscopy results for the identified ReOss^®^ defect: (**a**) area of interest 1; (**b**) area of interest 2.

**Figure 7 materials-13-02177-f007:**
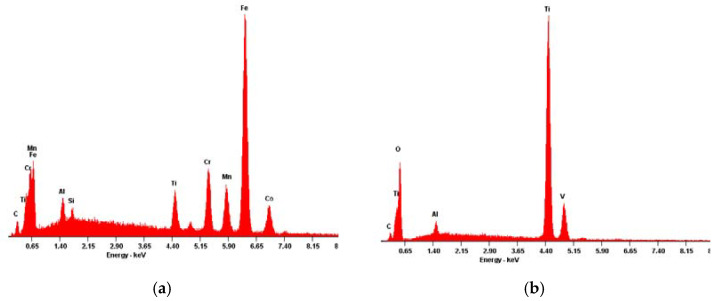
Energy dispersive X-ray spectroscopy results for the identified BTK defect: (**a**) area of interest 1; (**b**) area of interest 2.

**Figure 8 materials-13-02177-f008:**
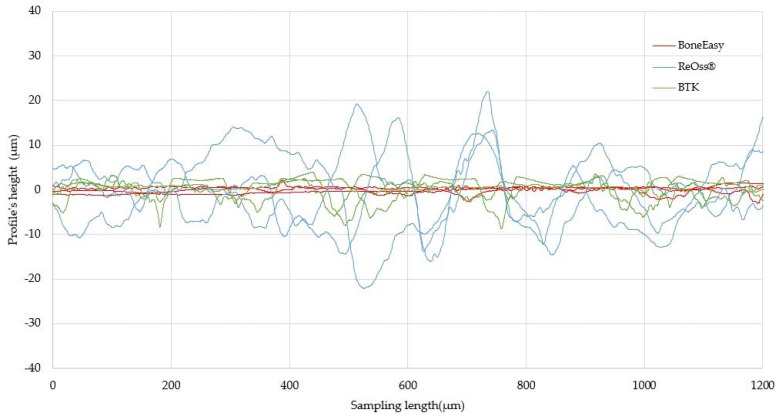
Plotted roughness profiles plotted from the profilometry analysis. In red, the BoneEasy’s triplicates’ profiles. In blue, the ReOss^®^’s triplicates’ profiles. In green, the BTK’s triplicates’ profiles.

**Table 1 materials-13-02177-t001:** Dimensions of the different samples’ surface features.

Sample	Pore Diameter	Peanut-Shape Length
BoneEasy	1.93 ± 0.11 mm	n/a
ReOss^®^	1.38 ± 0.03 mm	5.47 ± 0.10 mm
BTK	1.23 ± 0.04 mm	n/a

**Table 2 materials-13-02177-t002:** Roughness parameters for the different samples’ surfaces.

Sample	Ra	Rq
BoneEasy	0.61 ± 0.14 µm	0.73 ± 0.13 µm
ReOss^®^	6.59 ± 0.76 µm	8.39 ± 0.97 µm
BTK	1.63 ± 0.19 µm	2.08 ± 0.20 µm

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
