# Peer review of "Surface Comparison of Three Different Commercial Custom-Made Titanium Meshes Produced by SLM for Dental Applications"

_materials, 2020, doi:10.3390/ma13092177_

Round 1
Reviewer 1 Report
The authors did a study entitled “Surface comparison of three different commercial custom-made titanium meshes for bone regeneration produced by SLM”. The authors should do more expanded work and make major revision before publication.
- The title of this paper is “Surface comparison of three different commercial custom-made titanium meshes for bone regeneration produced by SLM”. However, none experiment for bone regeneration was presented in the paper. Thus, more biological work was suggested.
- The specific features of the three different commercial custom-made titanium meshes, such as shape, size, application and their influence for bone regeneration should be described in detailed.
- Different potential application of the bone implant required different conditions, such as osteogenesis, osteoclast, anticoagulation, angiogenesis. All these factors should be considered for the three different commercial custom-made titanium meshes.
- Several researches about interaction titanium and its biocompatibility are suggested, if possible.
Tailoring of the titanium surface by preparing cardiovascular endothelial extracellular matrix layer on the hyaluronic acid micro-pattern for improving biocompatibility. Colloids and Surfaces B: Biointerfaces, 2015, 128:201-210.
- The biocompatibility of the implants is very important, thus is suggested to supplement.
Author Response
Dear Reviewer,
Thanks for taking the time to review our manuscript and suggest how to improve our work. We are now submitting a manuscript that addresses the points you specifically raised among others we have considered in order to deliver what we think to be an improved version of our work.
Thanks a lot,
We are looking forward to your comments.
1. The title of this paper is “Surface comparison of three different commercial custom-made titanium meshes for bone regeneration produced by SLM”. However, none experiment for bone regeneration was presented in the paper. Thus, more biological work was suggested.
No biological work was conducted during this study as the main contribution of this paper is to compare the meshes’ physical and chemical characteristics alone, having the major focus fallen on the topographic characterization. Without a doubt, biological work could be conducted in future studies.
2. The specific features of the three different commercial custom-made titanium meshes, such as shape, size, application and their influence for bone regeneration should be described in detailed.
At a macroscopic level, the shape and size characterization is, probably, not significative as each mesh is custom made and their design is defined to fit individual defects. However, extended description of the surface patterns is now delivered, including the major features’ dimensions. The added data can be found introducing the Results section (lines 162-174); the corresponding measurement protocol was added to the section of materials and methods (lines (146-147). Further, in the Introduction section, the authors have introduced a more extensive analysis correlating the surface properties with the cellular response and, consequently the clinical outcome (lines 96-113).
3. Different potential application of the bone implant required different conditions, such as osteogenesis, osteoclast, anticoagulation, angiogenesis. All these factors should be considered for the three different commercial custom-made titanium meshes.
As no biological studies were intended to be conducted, no information regarding the angiogenic, osteogenic or anticoagulation potential of the mesh was gathered; the monitoring of such aspects can promote different potential applications and could be elucidated in future publications. However, it is important to note that the biological outcome of the process does not depend only on the above-mentioned properties but also on the correct diagnosis, clinical indications and the host characteristics. This addendum was added in the Introduction section (lines 130-134).
4. Several researches about interaction titanium and its biocompatibility are suggested, if possible.
5. The biocompatibility of the implants is very important, thus is suggested to supplement.
Regarding both the comment 4 and 5, the properties that are responsible for the titanium biocompatibility are now described. Further, the titanium relationship with the surrounding environment is mentioned along with its safety, particularly for dentistry use (lines 81-90).
Reviewer 2 Report
Recommendation:Major revision
Comments: This manuscript compared the surface morphology, chemical composition, and roughness of three different titanium meshes. The reported differences between the surfaces were ascribed to the divergent post-production superficial treatments applied. According to the surface roughness analysis, the Ra value of BoneEasy’s mesh was closer to the reported optimal roughness degree that enhances the osteoblasts’ affinity to the surface. However, the significance of this manuscript is unclear. It is very much like a research report rather than a high quality scientific paper. There are some important issues needed to be addressed. Thus, the reviewer can not recommend it to be published in this journal at the current stage.
The detailed comments are stated below:
(1) The major focus of this manuscript is the surface properties of titanium meshes, thus more description of the biological relevance of the surface properties can be given in the Introduction.
(2) The production technique of titanium occupies a great part in Introduction, but the relationship between different processes and the surface properties is rarely discussed. The authors should comment on how production technique influences the surface properties.
(3) Among all the different commercial titanium meshes, the reasons why these three meshes are selected can be clarified.
(4) The digital images of three meshes (Figure 1) could be pictured more elaborately by unifying the background.
(5) The scale of the Y-axis of the energy dispersive X-ray spectroscopy (Figures 3, 6, and 7) should be constant.
(6) Quantitative analysis and relevant reference of energy dispersive X-ray spectroscopy should be given to support the conjectures about the detected contaminations.
(7)Since EDS and XRD results show little difference in the composition among three meshes. This manuscript put particularly interest in roughness analysis. However, these results are not enough for a research paper . More comparison of the surface properties, such as wettability and hardness, are required.
(8) The last and most important is biocompatibility of three meshes should be evaluated. The surface properties highly affect the cell behaviors, thus studies about cell proliferation or spreading would be helpful to understand the difference among three meshes
Author Response
Dear Reviewer,
Thanks for taking the time to review our manuscript and suggest how to improve our work. We are now submitting a manuscript that addresses the points you specifically raised among others we have considered in order to deliver what we think to be an improved version of our work.
Thanks a lot,
We are looking forward to your comments.
(1) The major focus of this manuscript is the surface properties of titanium meshes, thus more description of the biological relevance of the surface properties can be given in the Introduction.
Little emphasis was given to the relationship between superficial features and biologic response in the Introduction section. The deficient analysis was replaced for a more complete interpretation that correlates the surface properties with the cellular response and, consequently the clinical outcome (lines 96-113).
(2) The production technique of titanium occupies a great part in Introduction, but the relationship between different processes and the surface properties is rarely discussed. The authors should comment on how production technique influences the surface properties.
The relationship between the SLM production method and the attained surface characteristics is now clarified, as well as the unavoidable demand of the superficial treatment application when resorting to this manufacturing process (lines122-129). Further, some of the usually applied surface treatments, among with the consequent surface alterations from it derived, are also reported (lines 114-122).
(3) Among all the different commercial titanium meshes, the reasons why these three meshes are selected can be clarified.
These particular meshes were selected as they are the ones available in the European market with a higher expression presence. Such information was added to the Materials and Methods section (lines143-145).
(4) The digital images of three meshes (Figure 1) could be pictured more elaborately by unifying the background.
The displayed images were captured with a regular mobile phone camera however, to avoid anticipated loss of the uncontaminated nature of the samples, they were photographed still inside the delivering package. Further, due to the meshes’ incompatible dimensions for some of the analysis and in order to efficiently manage the realization of all the tests, each mesh was sectioned, precluding the possibility of currently acquiring improved images.
(5) The scale of the Y-axis of the energy dispersive X-ray spectroscopy (Figures 3, 6, and 7) should be constant.
While the energy spectrum x-axis represents the X-rays energy, having as unit the electron-volt (eV), the y-axis registers the number of counts detected. The peak position leads to the element identification and the peak height to the quantification of each one’s concentration by a ratio analysis. In this way, comparisons between different EDS spectra are not adequate and, to
standardize the presented images, the y-axis from all the EDS results were excluded, following the strategy adopted from other authors in this journal published.
(6) Quantitative analysis and relevant reference of energy dispersive X-ray spectroscopy should be given to support the conjectures about the detected contaminations.
No quantitative analysis was preformed regarding the EDS results as their intend was to retrieve qualitative information on the overall meshes' composition so that the titanium presence could be verified. Supplementary, contaminations were found and the auhtors intent consisted only in an elementar identification of the contamination chemical identity in order to conjecture about its possible origin.
(7) Since EDS and XRD results show little difference in the composition among three meshes. This manuscript put particularly interest in roughness analysis. However, these results are not enough for a research paper. More comparison of the surface properties, such as wettability and hardness, are required.
Due to the limited access to characterization techniques, it was necessary to prioritize the ones that would deliver the most crucial information. For this study, the authors considered the most relevant characteristics that should be examined during an initial mesh comparison to be the topographic features and the surface roughness, as they have been able to deliver major characterization differences between the meshes.
(8) The last and most important is biocompatibility of three meshes should be evaluated. The surface properties highly affect the cell behaviors, thus studies about cell proliferation or spreading would be helpful to understand the difference among three meshes.
No biological work was conducted during this study as its main objective was to compare the meshes’ physical characteristics alone, having the major focus fallen on the superficial characterization. Without a doubt, biological work could be conducted in future studies.
Reviewer 3 Report
Surface comparison of three different commercial custom-made titanium meshes for bone regeneration produced by SLM is very interesting paper.
Some improvements are required.
Line 23 (and later Figure 2): XRD-Analysis was reported. What is result of XRD-analysis? Which compounds are present in your samples?
Line 81: Which production techniques?
Line 91: three different commercially available individualized titanium meshes produced by.....
Line 93: X-ray diffraction (XRD) was used to discern detailed information about the chemical structure of the materials. What is the chemical structure of samples . No quantitative XRD Rietvield analysis in text. Why?
Line 126: What are chemical compounds at the Figure 2.
Line 128. The name of Y-axis at Figures 3,6, 7 shall be written again in order to see it clear
Line 166: Considering the Yxoss CBR® defect (Figure 5a), two distinct types of infiltrations could be found. What is the basical chemical composition of three studied Meshes: Ti‐6Al‐4V ? What is role of presence of Si, Cr, V, Co?
Line 325: personalized production method (which methods?)
Line 336; mesh surface were found evidences of stainless-steel contamination. Is it confirmed by XRD or EDS-Analysis?
Conclusion: Which roughness value is permitted, and suitable for safe application and Long life duration of product.
General comments
1.Which oxides are problem in structure?
2. Can you control the roughness using some additives in synthesis?
3. What is the origin of carbon in the structure and his role?
4. What is with presence of TiC in structure?
Author Response
Dear Reviewer,
Thanks for taking the time to review our manuscript and suggest how to improve our work. We are now submitting a manuscript that addresses the points you specifically raised among others we have considered in order to deliver what we think to be an improved version of our work.
Thanks a lot,
We are looking forward to your comments.
Line 23 (and later Figure 2): XRD-Analysis was reported. What is result of XRD-analysis? Which compounds are present in your samples?
The XRD analysis results were only used to prove the chemical identity similarity between the three different meshes. To unveil the chemical elements present, it was decided to resort to a simple EDS assessment and the disclosed elements, either of the overall mesh and the contamination’s compositions, are presented in Figure 3, 6 and 7 and analysed in the Discussion section.
Line 81: Which production techniques?
The information that the implant properties must be controlled either when the implant is produced by the conventional methods or by rapid prototyping techniques has been clarified (91-92).
Line 91: three different commercially available individualized titanium meshes produced by.....
The method name, SLM, has been added (line 136).
Line 93: X-ray diffraction (XRD) was used to discern detailed information about the chemical structure of the materials. What is the chemical structure of samples . No quantitative XRD Rietvield analysis in text. Why?
No crystallographic phase composition analysis was desired but rather a qualitative information of the overall mesh composition. As this information could be obtained from the EDS characterization, no Rietvield analysis was conducted. In the future, for a more detailed characterization, further analysis such as a metallographic one should be performed.
Line 126: What are chemical compounds at the Figure 2.
The prime objective of the conduction of the XRD analysis was to possibly identify the meshes as pure Ti or as a Ti-alloy: the only possible material compositions as mentioned by the manufacturers. However, due to the issue mentioned on lines 288-299, relative to the impossibility of revealing such different compositions due to the reflections overlap, further exploration was not pursued. In this way, the only conclusion to deduce from this analysis is that the XRD spectra acquired indeed matches to pure Ti or a Ti-alloy material; information that has been clearly added to the Discussion section (lines 292-295).
Line 128. The name of Y-axis at Figures 3,6, 7 shall be written again in order to see it clear
While the energy spectrum x-axis represents the X-rays energy, having as unit the electron-volt (eV), the y-axis registers the number of counts detected. The peak position leads to the element identification and the peak height to the quantification of each one’s concentration by a ratio analysis. In this way, comparisons between different EDS spectra are not adequate and, to
standardize the presented images, the y-axis from all the EDS results were excluded, following the strategy adopted from other authors in this journal published.
Line 166: Considering the Yxoss CBR® defect (Figure 5a), two distinct types of infiltrations could be found. What is the basical chemical composition of three studied Meshes: Ti‐6Al‐4V ? What is role of presence of Si, Cr, V, Co?
The basic chemical identity of the meshes, that allowed the future finding of contaminations is presented in Figure 3. The authors decided not to largely expand on the contaminations detailed composition as they are not considered a crucial information and their presence is not desired. The main focus of these results’ analysis fell on the conjecture about the contaminations origin and possible implications for its physical properties and future clinical use.
Line 325: personalized production method (which methods?)
With the “personalized production methods” statement the authors intend to mention the ones that are based in modelling and 3D printing, resulting in perfect fits between the construct and the patient’s defects, as specified in lines 384-386.
Line 336; mesh surface were found evidences of stainless-steel contamination. Is it confirmed by XRD or EDS-Analysis?
The evidences of stainless-steel contaminations found on the BTK mesh surface were revealed by EDS analysis. However, since that information was already provided both in the Results and Discussion section, it was decided to not include it again on the conclusion section to maintain the coherence with the other analysis results and facilitate the conclusions interpretation by the reader.
Conclusion: Which roughness value is permitted, and suitable for safe application and Long life duration of product.
For the development of this work the considered ideal Ra roughness value for the meshes’ surface was 0.5 µm, reported to promote the greatest affinity between the osteoblasts and the mesh surface. This value has been reported again to the Conclusions section from the readers’ retention (line 391). Even though it is not expected to deliver safer clinical applications outcomes when comparing to the other meshes characteristics, the BoneEasy mesh is the one that closer gets to this reported optimal roughness degree.
Round 2
Reviewer 1 Report
Although the author has made some improvement on their work, it is due to their title“... for bone regeneration”. I still insist that they complement biological experiments. Or they can change the title of the article.
Author Response
Dear Reviewer,
Again, thank you so much for your feedback.
As suggested, the work’s title as been changed to “Surface comparison of three different commercial custom-made titanium meshes produced by SLM for dental applications”
Best regards,
The authors
Reviewer 2 Report
It is well revised.
Author Response
Dear Reviewer,
Thank you so much for all your comments that allowed the delivery of an improved work.
Best regards,
The authors
Reviewer 3 Report
Thank you for your answer and improvement. No additional questions.
Author Response

(The authors gave the same response as above.)

Round 3
Reviewer 1 Report
Although there are many places to modify the article, the reviewer agrees to publish it in "Materials" in the current state.Author Response
Dear Reviewer,
Thank you so much for all your comments that allowed the delivery of an improved work.
Best regards,
The authors